# Possible Involvement of Circulating Immune Complex Containing IgG4 in the Pathogenesis of IgG4-Related Disease Complicated by Hypocomplementemia: A Case Report

**DOI:** 10.3390/ijms262110687

**Published:** 2025-11-03

**Authors:** Takahiro Uchida, Yuka Miyake, Sachiko Iwama, Ken Aoki, Dan Inoue, Muneharu Yamada, Takashi Oda

**Affiliations:** Department of Nephrology and Blood Purification, Kidney Disease Center, Tokyo Medical University Hachioji Medical Center, 1163 Tatemachi, Hachioji, Tokyo 193-0998, Japan; tu0509@tokyo-med.ac.jp (T.U.); miyake.yuka.8j@tokyo-med.ac.jp (Y.M.);

**Keywords:** complement activation pathway, hypocomplementemia, IgG4, IgG4-related disease, immune complex, tubulointerstitial nephritis

## Abstract

Studies examining IgG subclasses within circulating immune complexes (CICs) in patients with IgG4-related disease remain scarce. A Japanese man in his 50s with a history of diabetes mellitus and chronic pancreatitis was referred to our department because of an increase in serum creatinine levels. Serum IgG and IgG4 levels were markedly high, accompanied by eosinophilia and elevated serum IgE levels. C3 hypocomplementemia and an increase in CICs were also noted, and imaging revealed swollen mediastinal lymph nodes. Renal biopsy revealed extensive tubulointerstitial nephritis with numerous IgG4-positive plasma cells and dense interstitial fibrosis. The patient was diagnosed with IgG4-related disease, and glucocorticoid therapy was initiated; renal function, serological abnormalities, and swelling of the mediastinal lymph nodes improved. Subsequent analyses revealed that the patient’s CICs mainly comprised IgG4 and that there was tubular deposition of complement components C1q, C4d, C3, and C5b-9 in the renal biopsy tissue, suggesting that immune complexes containing IgG4 activated the complement pathway in circulation and locally in the kidneys. Hypocomplementemia and CICs are observed in a subset of patients with IgG4-related diseases; however, the underlying mechanisms remain unclear. Further accumulation of IgG4-related disease cases is required to evaluate the possibility of IgG4-mediated complement activation.

## 1. Introduction

IgG4-related disease (IgG4-RD) is a systemic fibroinflammatory condition characterized by increased serum IgG4 concentrations and dense lymphoplasmacytic infiltration that is accompanied by storiform fibrosis [1]. This disease also has the characteristics of dissemination in time and space and can affect almost all organs. Although the prognosis of patients with IgG4-RD is generally regarded as favorable because of good responses to glucocorticoid therapy, there have been several cases of patients presenting with treatment-refractory clinical courses [2,3,4]. To date, definitive therapeutic regimens for IgG4-RD have not been established, as the disease pathogenesis remains to be fully elucidated.

Both innate and adaptive immune responses are involved in the pathogenesis of IgG4-RD [5], but the activation of the complement system has recently attracted attention. Not a few patients with IgG4-RD present with hypocomplementemia [6], which is reportedly associated with a high risk of disease relapse [7]. The prevalence of hypocomplementemia is further increased in patients with IgG4-related kidney disease (IgG4-RKD), of which tubulointerstitial nephritis (TIN) with marked infiltration of IgG4-positive plasma cells is a characteristic renal histopathological finding [8]. Hypocomplementemia in IgG4-RKD is reportedly associated with the severity of interstitial inflammation and fibrosis [9]. However, IgG4 is the least abundant IgG subclass in circulation and is considered the least pro-inflammatory, in part because it poorly activates the classical complement pathway. Therefore, the mechanisms underlying hypocomplementemia in patients with IgG4-RD remain unclear.

We recently reported a case of IgG4-RKD with marked hypocomplementemia, in which circulating immune complexes (CICs), predominantly of the IgG3 subclass, were thought to play crucial roles in pathogenesis through complement activation via the classical pathway despite only a slight increase in the total serum levels of IgG3 (Miyake et al. Kidney Med. in press). However, studies examining IgG subclasses within CICs of patients with IgG4-RD remain scarce. Here, we present a case of IgG4-RD complicated by hypocomplementemia, in which IgG4-dominant CICs were observed.

## 2. Case Report

A Japanese man in his 50s was referred to our department because of an increase in serum creatinine levels over the past 6 months (from 0.9 mg/dL to 1.8 mg/dL). The laboratory data are summarized in Table 1. Urinalysis results were unremarkable on dipstick tests, including occult blood and protein tests, whereas markers of renal tubular injury were elevated. Marked increases in the serum IgG and IgG4 levels (without evidence of monoclonal protein), eosinophilia, and elevated serum IgE levels were noted. Serum analysis of IgG subclasses revealed a marked increase in IgG1 and IgG2 levels, whereas IgG3 levels remained within the normal range (Table 2). Furthermore, C3 hypocomplementemia and an increase in CICs were observed using a monoclonal rheumatoid factor (mRF)-based assay (SRL, Inc., Tokyo, Japan). Computed tomography revealed swelling of the mediastinal lymph nodes and bilateral kidneys; however, his vital signs were within normal ranges, and physical examination findings were unremarkable, with no rash or superficial lymphadenopathy. He had a history of diabetes mellitus, which was well controlled with insulin treatment, and chronic pancreatitis of unknown origin.

Light microscopy revealed extensive TIN, with dense interstitial fibrosis surrounding the infiltrating cells in renal biopsy specimens (Figure 1A). Immunostaining revealed numerous CD138-positive plasma cells in the tubulointerstitial area and >50 IgG4-positive cells per high-power field, with an IgG4/IgG-positive cell ratio of approximately 70% (Figure 1B). None of the 16 glomeruli were globally or segmentally sclerotic, and only minor abnormalities were observed (Figure 1C). Immunofluorescence staining of fresh frozen tissue sections revealed no deposition of immunoglobulins or complement components in the glomeruli (Figure 1D), and electron microscopy revealed no electron-dense deposits in either the glomeruli or tubules (Figure 1E). In contrast, deposition of complement components C4d, C3, and C5b-9 was observed in the tubules (Figure 1F). Although C1q deposition in the tubules was weak (Figure 1G), indirect immunostaining for immune complexes (ICs) using a mouse anti-mRF antibody (provided by Nissui Pharmaceutical Co., Ltd., Tokyo, Japan) as the primary antibody and Alexa Fluor 488-conjugated goat anti-mouse IgG as the secondary antibody showed positive staining in some tubules (Figure 1G). Furthermore, double immunofluorescence staining for mRF and C4d demonstrated their co-localization in the tubules (Figure 1H). There findings strongly suggested the presence of ICs and activation of the classical complement pathway. There were no signs of diabetic nephropathy in renal biopsy tissue specimens. Endobronchial ultrasound-guided transbronchial needle aspiration of the mediastinal lymph nodes showed infiltration of CD138-positive plasma cells but ruled out the possibility of malignancies, such as lymphoma.

An enzyme-linked immunosorbent assay using an mRF assay plate (Nissui Pharmaceutical Co., Ltd.) showed that IgG4 was the only IgG subclass detected within the CIC of the patient (Table 2). To further confirm this finding, Western blot analysis [10] of CICs isolated from the patient’s serum using a protein G agarose-resin column (Thermo Fisher Scientific, Waltham, MA, USA) conjugated with a mouse anti-mRF antibody (provided by Nissui Pharmaceutical Co., Ltd.) further confirmed the predominance of IgG4 among the IgG subclasses (Figure 2).

The diagnosis of IgG4-RD was made based on the comprehensive diagnostic criteria [11]. Glucocorticoid therapy consisting of prednisone at 30 mg/day (0.6 mg/kg/day) was initiated, which improved the patient’s renal function, serological abnormalities, and swelling of the mediastinal lymph nodes. The prednisone dose was tapered to 5 mg/day, and the patient maintained clinical improvement during the 4-year follow-up period. The serum creatinine level was approximately 1.0 mg/dL, and hypocomplementemia was no longer present.

## 3. Discussion

In this case, the renal pathology showed extensive TIN-dense interstitial fibrosis, characteristic of IgG4-RKD [8]. The pathological finding of the mediastinal lymph node seemed to be compatible with that of IgG4-RD; his past history of chronic pancreatitis could also be a sign of IgG4-RD. However, hypocomplementemia and an increase in CICs were observed, and the CICs mainly consisted of the IgG4 subclass, which was a notable finding demonstrated using enzyme-linked immunosorbent assay and Western blotting (Table 2 and Figure 2). Furthermore, deposition of complement components C1q (weak), C4d, C3, and C5b-9 was observed in the tubules of renal biopsy tissues (Figure 1F,G), and immunofluorescent staining using an anti-mRF antibody (suspected to bind with immune complexes) showed positive results in some tubules (Figure 1G). These findings indicated that ICs containing IgG4 activated the classical complement pathway not only in circulation but also locally in the kidneys.

However, hypocomplementemia in IgG4-RD is reportedly associated with elevated serum concentrations of IgG subclasses other than IgG4 [12,13]. CICs, including IgG subclasses except for IgG4, have been suggested to induce the activation of the classical complement pathway [14,15], although there are methodological differences between studies in terms of CIC detection. For example, IgG1-type CICs were suggested to activate the classical complement pathway in autoimmune pancreatitis, the most common manifestation of IgG4-RD, as levels of CIC and serum IgG1 were reportedly elevated and correlated with hypocomplementemia [16]. In contrast, in our recently reported case of IgG4-RKD (Miyake et al. Kidney Med. in press), elevated levels of IgG3-dominant CICs and IgG3-positive large materials and deposition of complement components C1q, C4d, C3, and C5b-9 in the renal tubules were demonstrated, suggesting a potential role of IgG3-type ICs in the activation of the classical complement pathway. Importantly, IgG1 and IgG3 are well-known activators of the classical complement pathway, whereas IgG4 is considered the least pro-inflammatory IgG subclass because of its low binding capacity to complement C1q and Fcγ receptors [15]. Thus, the mechanisms by which IgG4-dominant CICs activated the complement pathway in our case should be discussed thoroughly.

In this regard, protein glycosylation, acknowledged as one of the major posttranslational modifications, might be related to complement activation in IgG4-RD; a previous report showed significant differences in the fucosylation of IgG4 glycans between IgG4-RD with and without hypocomplementemia, suggesting that IgG4 fucosylation change is related to complement activation in IgG4-RD [15]. Furthermore, the researchers showed that CICs from patients with IgG4-RD accompanied by hypocomplementemia, which were precipitated using polyethylene glycol, contained IgG4 (and IgM) and activated the complement system [6]. In contrast, IgG4 has been reported to activate the classical complement pathway only at high antigen and high antibody concentrations, thereby indicating that IgG4-mediated complement activation is possible but unlikely [17]. Further studies with large patient cohort and the accumulation of fundamental in vitro data are required to clarify the role of IgG4 in complement activation in IgG4-RD.

The prognosis of patients with IgG4-RD is generally good owing to favorable responses to glucocorticoid therapy, and hypocomplementemia in IgG4-RKD is reportedly associated with even better response to treatment [2]. However, the etiological significance of hypocomplementemia in IgG4-RD remains unclear, as several studies have reported conflicting results. For example, hypocomplementemia has been associated with renal interstitial inflammation and fibrosis, as well as a high risk of disease relapse [7,9]. Furthermore, there have been reported cases of IgG4-RKD with hypocomplementemia progressing to end-stage kidney disease, including one case showing deposition of IgG2 and IgG4, but not IgG1 or IgG3, in the tubulointerstitial area [3,4].

## 4. Conclusions

In this case of IgG4-RKD complicated by hypocomplementemia, IgG4-dominant CICs were observed. We propose that the IgG subclass composition in CICs of patients with IgG4-RD should be investigated in future studies using the same methodology at various laboratory facilities, which could further unveil the mechanisms underlying hypocomplementemia in this disease.

## Figures and Tables

**Figure 1 ijms-26-10687-f001:**
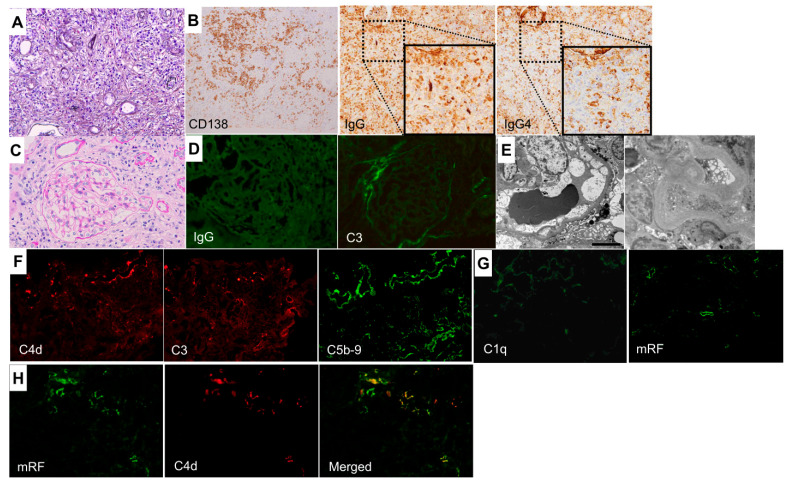
Renal histopathological findings. (**A**) Marked interstitial lymphoplasmacytic infiltration and surrounding fibrosis, forming a characteristic a bird’s eye pattern (periodic acid-methenamine-silver stain). Original magnification: ×200 (**B**) The infiltrating cells are predominantly IgG4-positive plasma cells (left: immunoperoxidase staining of formalin-fixed paraffin-embedded tissue sections for CD138; middle: immunoperoxidase staining for IgG; right: immunoperoxidase staining for IgG4). Original magnifications: ×200 (**C**) The glomeruli appear normal, without evidence of proliferative changes or thickening of the glomerular capillary walls (periodic acid–Schiff stain). Original magnification: ×400 (**D**) Immunofluorescence staining of fresh-frozen tissue sections reveals no deposition of immunoglobulins or complements in the glomeruli. Original magnifications: ×400 (**E**) Electron microscopy reveals no electron-dense deposits in either the glomeruli (left) or the tubules (right). (**F**) Deposits of complement components C4d, C3, and C5b-9 in the tubular basement membrane are observed by immunofluorescence staining of fresh frozen tissue sections. Original magnifications: ×200 (**G**) Immunofluorescence staining using fresh-frozen tissue sections shows weak C1q deposition in the tubular basement membrane (left). Staining using a mouse anti-mRF antibody (provided by Nissui Pharmaceutical Co., Ltd., Tokyo, Japan), followed by Alexa Fluor 488-conjugated goat anti-mouse IgG, shows positive staining in some tubules (right). Original magnifications: ×200 (**H**) Double immunofluorescence staining for mRF (left) and C4d (middle) using fresh-frozen tissue sections. A merged image is shown in the right panel. Original magnifications: ×200.

**Figure 2 ijms-26-10687-f002:**
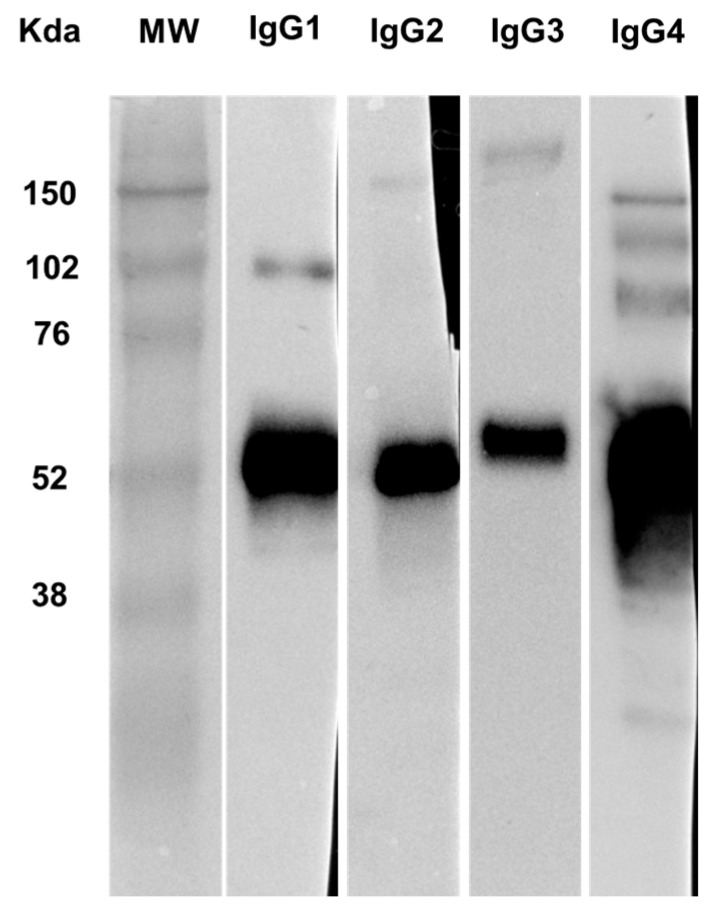
Western blot analysis of IgG subclasses in CICs isolated from the patient’s serum. A mouse anti-mRF antibody (provided by Nissui Pharmaceutical Co., Ltd., Tokyo, Japan) was conjugated to a protein G agarose column (Thermo Fisher Scientific, Waltham, MA, USA), which was used to isolate CICs from the patient’s serum. The extracted CICs were subjected to SDS-PAGE and immunoblotted with rabbit anti-human IgG1–IgG4 antibodies (Abcam, Cambridge, UK) and horseradish peroxidase-conjugated donkey anti-rabbit IgG antibody (Cytiva, San Diego, CA, USA). After reacting with chemiluminescent reagents, all blots were exposed at a standardized time (1 s). The major band at approximately 50–60 kDa appears to correspond to the IgG heavy chain monomer. The weaker, multiple bands of larger sizes observed mainly in the IgG4 and IgG1 blots likely represent other IgG components, including heavy chain–light chain complex (expected molecular weight around 80 kDa), heavy chain dimer (expected molecular weight 100–115 kDa), and full-length IgG dimer (expected molecular weight around 150–170 kDa). IgG4 displayed the strongest band. MW indicates the molecular weight marker.

**Table 1 ijms-26-10687-t001:** Initial laboratory data of the patient.

**Urinalysis**		TP/Alb	11.4/2.8 g/dL
Occult blood	-	Na/K/Cl	136/4.5/102 mmol/L
Protein	+	Ca/P	9.0/3.2 mg/dL
Red blood cell	1–4/HPF	Amylase	138 IU/L
White blood cell	1–4/HPF	Hemoglobin A1c	6.3%
Proteinuria	0.81 g/gCr (Alb: 14.6%)		
NAG	12.2 IU/L (1.0–4.2)	**Serology**	
β2-microglobulin	15,098 μg/L (0–287)	IgG/A/M	6060/74/37 mg/dL
		IgG4	4030 mg/dL (11–121)
**Complete blood count**		IgE	704 IU/L (0–172)
White blood cells	5790/µL	CH50	37.1 U/mL
Neutrophil/Lymphocyte/Eosinophil	44.8/21.6/23.1%	Complement C3/C4	67.1/20.4 mg/dL
Hemoglobin	10.8 g/dL	CIC (mRF)	23.5 µg/mL (0–4.1)
Platelet	239,000/µL	CRP	0.12 mg/dL
		sIL-2R	1553 U/mL
**Biochemistry**		ANA	<40
Blood urea nitrogen	33.9 mg/dL	MPO-ANCA	0.10 U/mL
Creatinine	1.80 mg/dL	PR3-ANCA	0.50 U/mL
eGFR_creat_	32.3 mL/min/1.73 m^2^		

Alb, albumin; ANA, antinuclear antibody; ANCA, antineutrophil cytoplasmic antibody; CIC, circulating immune complex; CRP, C-reactive protein; eGFR, estimated glomerular filtration rate; HPF, high-power field; MPO, myeloperoxidase; mRF, monoclonal rheumatoid factor; NAG, N-acetyl-β-glucosaminidase; PR3, proteinase 3; sIL-2R, soluble interleukin-2 receptor; TP, total protein. Parentheses denote normal ranges.

**Table 2 ijms-26-10687-t002:** Analyses of the patient’s IgG subclass in serum and circulating immune complexes.

	IgG1	IgG2	IgG3	IgG4
**Serum level (mg/dL)**	2530 (351–962)	2770 (239–838)	87.0 (8.5–140)	3880 (4.5–117)
**Circulating immune complex level**	Below DL	Below DL	Below DL	0.107

Parentheses denote normal ranges. The upper section of the table shows the results for IgG subclasses in the serum of the patient, which were evaluated using the immunoturbidimetric method in the clinical laboratory testing industry (SRL Inc., Hachioji, Tokyo, Japan). Prominently increased serum IgG1, IgG2, and IgG4 levels and normal IgG3 levels were observed. The lower section of the table shows the levels of IgG subclasses in circulating immune complexes (CICs), as determined by an enzyme-linked immunosorbent assay using an mRF antibody-coated plate (Nissui Pharmaceutical Co., Ltd., Tokyo, Japan). The patient’s serum samples were diluted 1:100 with dilution buffer and reacted with the plate in accordance with the manufacturer’s instructions to bind the CICs to the plate. After washing, the bound CICs were incubated with rabbit monoclonal antibodies against each IgG subclass (all purchased from Abcam, Cambridge, UK) and with alkaline phosphatase-conjugated goat anti-rabbit IgG (Sigma-Aldrich Co. LLC, St. Louis, MO, USA). The enzymatic reaction was performed using p-nitrophenylphosphate as the chromogenic substrate, and the optical density (OD) was measured at 405 nm. Dilution buffer, added to the plates in duplicate, was processed in a similar manner and was regarded as the negative control. The values calculated by subtracting the mean OD of the negative control from the OD obtained from the patient’s serum were interpreted as the relative amounts of each IgG subclass contained in the CICs and are shown in the table. “Below DL” indicates that the OD value was below the detection limit.

## Data Availability

The data presented in this study are available from the corresponding author upon request. The data is not publicly available due to ethical and privacy limitations.

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
