# Peer review of "Possible Involvement of Circulating Immune Complex Containing IgG4 in the Pathogenesis of IgG4-Related Disease Complicated by Hypocomplementemia: A Case Report"

_ijms, 2025, doi:10.3390/ijms262110687_

Round 1

Reviewer 1 Report

Comments and Suggestions for Authors

The comments follow throughout the attached document.

Reviewer 2 Report

Comments and Suggestions for Authors

In this interesting case authors present a male patient with IgG4 related disease where they examined the nature of circulating immune complexes and whether these contain IgG4 subclass immunoglobulin. Although the case is interesting there are certain features that need clarification.

Please provide high resolution IHC biopsy pictures. IgG IHC is not clear.

Please also provide immunofluorescence picture for IgG staining. In picture D, IgG seems to be negative in the peri-glomerular interstitial area. Please comment.

Please explain why mRF is only presented partially in two tubules in G picture and not in interstitial area as well or in other tubules.

In figure 2, WB shows multiple positive bands from 102 to 150 KD, especially in IgG1 and IgG4 columns. Please elaborate.

Round 2

Reviewer 1 Report

Comments and Suggestions for Authors

Most of the changes have been made and in my opinion the paper can be published.

Author Response

We sincerely thank you for your positive comments on our manuscript.

Reviewer 2 Report

Comments and Suggestions for Authors

I would like to thank the authors for their detailed respond to my original comments. However, in my opinion, immunofluorescence pictures potentially show only co-localization for C4d and C5b-9 but not with anti-mRF. In any case only confocal microscopy techniques would be appropriate for such an investigation or use of consecutive slides for each staining (in my opinion the last option was used for C4d and C5b-9 stainings). Thus, in my opinion co-localization of anti-mRF and C4d or C5b-9 is not documented.

Round 3

Reviewer 2 Report

Comments and Suggestions for Authors

Authors have adequately responded to my comments.